# Deficiency of Crif1 in hair follicle stem cells retards hair growth cycle in adult mice

**Jung-Min Shin[1], Jung-Woo Ko[1], Chong-Won Choi[1], Young Lee[1,2], Young-Joon Seo[1,2], Jeung-Hoon Lee[1,2], Chang-Deok Kim** [1,2]*

1 Department of Dermatology, School of Medicine, Chungnam National University, Daejeon, Korea,
2 Department of Medical Science, School of Medicine, Chungnam National University, Daejeon, Korea

* cdkimd@cnu.ac.kr

## Abstract

Hair growth is the cyclically regulated process that is characterized by growing phase (anagen), regression phase (catagen) and resting phase (telogen). Hair follicle stem cells (HFSCs) play pivotal role in the control of hair growth cycle. It has been notified that stem cells have the distinguished metabolic signature compared to differentiated cells, such as the preference to glycolysis rather than mitochondrial respiration. Crif1 is a mitochondrial protein that regulates the synthesis and insertion of oxidative phosphorylation (OXPHOS) polypeptides to inner membrane of mitochondria. Several studies demonstrate that tissue-specific knockout of Crif1 leads to mitochondrial dysfunction. In this study, we investigated the effect of mitochondrial dysfunction in terms of Crif1 deficiency on the hair growth cycle of adult mice. We created two kinds of inducible conditional knockout (icKO) mice. In epidermal specific icKO mice (Crif1 K14icKO), hair growth cycle was significantly retarded compared to wild type mice. Similarly, HFSC specific icKO mice (Crif1 K15icKO) showed significant retardation of hair growth cycle in depilation-induced anagen model. Interestingly, flow cytometry revealed that HFSC populations were maintained in Crif1 K15icKO mice. These results suggest that mitochondrial function in HFSCs is important for the progression of hair growth cycle, but not for maintenance of HFSCs.

## Introduction

Hair follicle is a self-renewing organ showing cyclicity composed of anagen (growing phase), catagen (regression phase) and telogen (resting phase). The cyclicity of hair follicle indicates the presence of its own stem cells. After discovery of hair follicle stem cells (HFSCs) resided in bulge region, many studies demonstrate that HFSCs play an important role in hair growth cycle as well as epidermal tissue regeneration [1].

Stem cells have a specific metabolic signature that is distinguished from the differentiated somatic cells. For example, embryonic stem cells (ESCs) prefer the glycolysis as a major source of energy production. However, during the differentiation process, they undergo the metabolic shift from glycolysis to mitochondrial respiration [2]. Supportedly, in induced pluripotent stem cells (iPSCs), cellular reprogramming process resets mitochondrial function to an

**Data Availability Statement:** All relevant data are within the manuscript and its Supporting Information files.

**Funding:** This study was supported by a grant from the National Research Foundation of Korea

(NRF-2017R1A2B4008810). The funders had no role in study design, data collection and analysis, decision to publish, or preparation of the manuscript.

**Competing interests:** The authors have declared that no competing interests exist.

immature level similar to ESCs [3,4]. In other example, during the differentiation of HFSCs, mitochondria elongate with more cristae and show higher activity, accompanying with activated aerobic respiration [5]. These results indicate that mitochondrial function is important in determination of stem cell characteristics. Despite the importance of mitochondrial function, it is poorly studied whether mitochondrial dysfunction affects hair growth cycle in adult mice.

Crif1 is a mitochondrial protein (S1 Fig), which regulates the synthesis and insertion of oxidative phosphorylation (OXPHOS) polypeptides by interacting with mitoribosomal large subunit [6]. Several tissue-specific conditional knockout mice show that Crif1 is essential for mitochondrial function in various tissues such as brain, heart and adipose tissues [7,8]. We have recently reported that targeted deletion of Crif1 in epidermis using K14-Cre results in decrease of keratinocyte proliferation and differentiation due to mitochondrial dysfunction. We have also observed that hair morphogenesis is severely hampered in epidermal specific Crif1 knockout newborn mice (K14-Cre;Crif1$^{fl/fl}$) [9].

In this study, we established inducible conditional knockout (icKO) mice to investigate the effect of mitochondrial dysfunction on hair growth of adult mice. We generated Crif1 icKO mice using a keratin-14 (K14) promoter for epidermal and hair follicle keratinocytes (K14-CreERT;Crif1$^{fl/fl}$) and a keratin-15 (K15) promoter for HFSCs (K15-CrePR;Crif1$^{fl/fl}$). Hair growth was delayed by Crif1 loss in epidermis and HFSCs. Our data suggest that mitochondrial function is important for the progression of hair growth cycle.

## Materials and methods

### Mice

Crif1$^{fl/fl}$ mice were generated as previously described [10], and were crossed with KRT14-CreERT2 (The Jackson Laboratory, Bar Harbor, Maine) or KRT15-CrePR mice (generously provided by Dr. George Cotsarelis, Philadelphia, PA). For induction of knockout, mice were shaved at P21 and topically treated with 4-hydroxy tamoxifen (4-OHT) or RU486 (Cayman, Item No. 10006317, 200 μl of 5 mg/ml in acetone) during first telogen (P22-P26). And hair growth was examined around P35-P44.

For depilation-induced anagen, RU486 (200 μl of 5 mg/ml in corn oil) was intraperitoneally injected daily for 10 times during 2 weeks (P35-P48). The left back skin were depilated at P49 and right back skin was depilated at P56 using hair removal wax. Hair growth was examined at P63.

All experiments were performed in accordance with institutional guidelines and approved by Chungnam National University institutional animal care and use committee (IRB CNU-00654). Mice were maintained in conventional condition with food and water *ad libitum* and monitored daily to minimize animal suffering. Mice were sacrificed using $CO_2$ gas.

### Quantitative reverse transcription-polymerase chain reaction (qRT-PCR)

The back skin tissues were dissected from mice, and then floated on trypsin solution (Thermo Scientific, Rockford, IL) at 4°C overnight. Epidermis was separated from dermis using the fine forcep. Total RNA was isolated using RNA mini kit (Ambion, Austin, TX) and reverse-transcribed with moloney-murine leukaemia virus (M-MLV) reverse transcriptase (ELPIS Biotech, Daejeon, Korea). qRT-PCR was performed on Applied Biosystems StepOne with SYBR Green real-time PCR master mix (Applied Biosystems, Foster City, CA) according to the manufacture's protocol. The relative expression levels of mRNA were determined by the comparative Ct method. The primer sequences were as follows: Crif1 (`5'-GCGAAAGCAGAAGCGAGAA C-3'`, `5'-GGCCCTCCGCTCCTTGT-3'`), Actin (`5'-CGATGCCCTGAGGCTCTTT-3'`, `5'-TGGATGCCACAGGATTCCA-3'`).

## Histology and immunostaining

Tissue samples were fixed with 10% formaldehyde, embedded in paraffin, and cut into 4-μm-thick sections. Sections were deparaffinized in xylene and then rehydrated by alcohol series. To examine the histology, sections were stained with hematoxylin and eosin (H&E). For immunohistochemistry, sections were first treated with 3% $H_2O_2$ to block the endogenous peroxidase, then incubated with IHC protein block solution (DAKO, Carpinteria, CA). Sections were then reacted with primary antibody at 4°C for overnight, then sequentially reacted with horseradish peroxidase-conjugated secondary antibody (DAKO). After washing, sections were incubated with diaminobenzidine tetrachloride solution and counterstained with Mayer's hematoxylin. For double immunofluorescence staining, sections were incubated with primary antibodies, then incubated with fluorescence-conjugated secondary antibodies (Abcam, Cambridge, UK). Immunofluorescence signal was detected under a fluorescence microscope (Olympus Corporation, Tokyo, Japan). The following primary antibodies were used: Crif1 (Santa Cruz Biotechnology, Santa Cruz, CA), MTCO1 (Abcam), K15 (Abcam), K5 (Santa Cruz Biotechnology, Santa Cruz, CA), AE15 (Thermo Fisher Scientific), AE13 (Abcam), Lgr5 (Thermo Fisher Scientific) and Ki67 (Vector Laboratories, Burlingame, CA).

## Flow cytometry

Preparation of bulge cells and total epidermal keratinocytes from adult mouse back skins was described previously [11]. Briefly, epidermis was separated from dermis after trypsin treatment. The collected epidermis was vigorously pipetted and filtered through a cell strainer. After centrifugation, cells were suspended in PBS and stained with antibodies for hair follicle stem cell markers: anti-integrin-α6 (CD49f) antibody directly coupled to PE (BD Biosciences, San Jose, CA), anti-CD34 antibody directly coupled to FITC (BD Biosciences). After washing with PBS, cells were analyzed using FACScaliber (BD Biosciences).

## Statistical analysis

All experiments were repeated at least three times with separate batches. Data were evaluated statistically using Mann-Whitney test. Statistical significance was set at $p < 0.05$.

# Results

## Delayed hair growth cycle by Crif1 loss in epidermis

We created the epidermal specific inducible conditional knockout (icKO) mice because that K14-Cre;Crif1[fl/fl] mice died within a week after birth. [9] We bred K14-CreERT transgenic mice with Crif1[fl/fl] mice. The resulting K14-CreERT;Crif1[fl/fl] mice (Crif1 K14icKO) and littermate Crif1[fl/fl] (WT) mice were topically treated with tamoxifen from P21 for 5 days. At P35, targeted deletion of Crif1 was verified by quantitative-PCR using epidermal lysates (Fig 1A). In immunohistochemistry, the expression of Crif1 was markedly reduced in Crif1 K14icKO mice. Together, MTCO1 (an mtDNA-encoded subunit of OXPHOS) was remarkably reduced in Crif1 K14icKO mice, indicating that mitochondrial dysfunction was successfully achieved by Crif1 loss (Fig 1B). After induction of Crif1 loss by tamoxifen treatment, Crif1 K14icKO showed delayed anagen induction compared to WT mice. At P35, the back skin of WT mice was covered with newly grown black hairs (anagen appearance), while the back skin of Crif1 K14icKO mice showed pinkish telogen appearance (Fig 1C). Histological analysis confirmed that hair follicles were at anagen phase in WT mice while at telogen phase in Crif1 K14icKO mice, evidenced by hair follicle length (Fig 1D). To examine the proliferative status of hair follicle cells, we performed immunohistochemical staining using Ki67 antibody. The Ki67

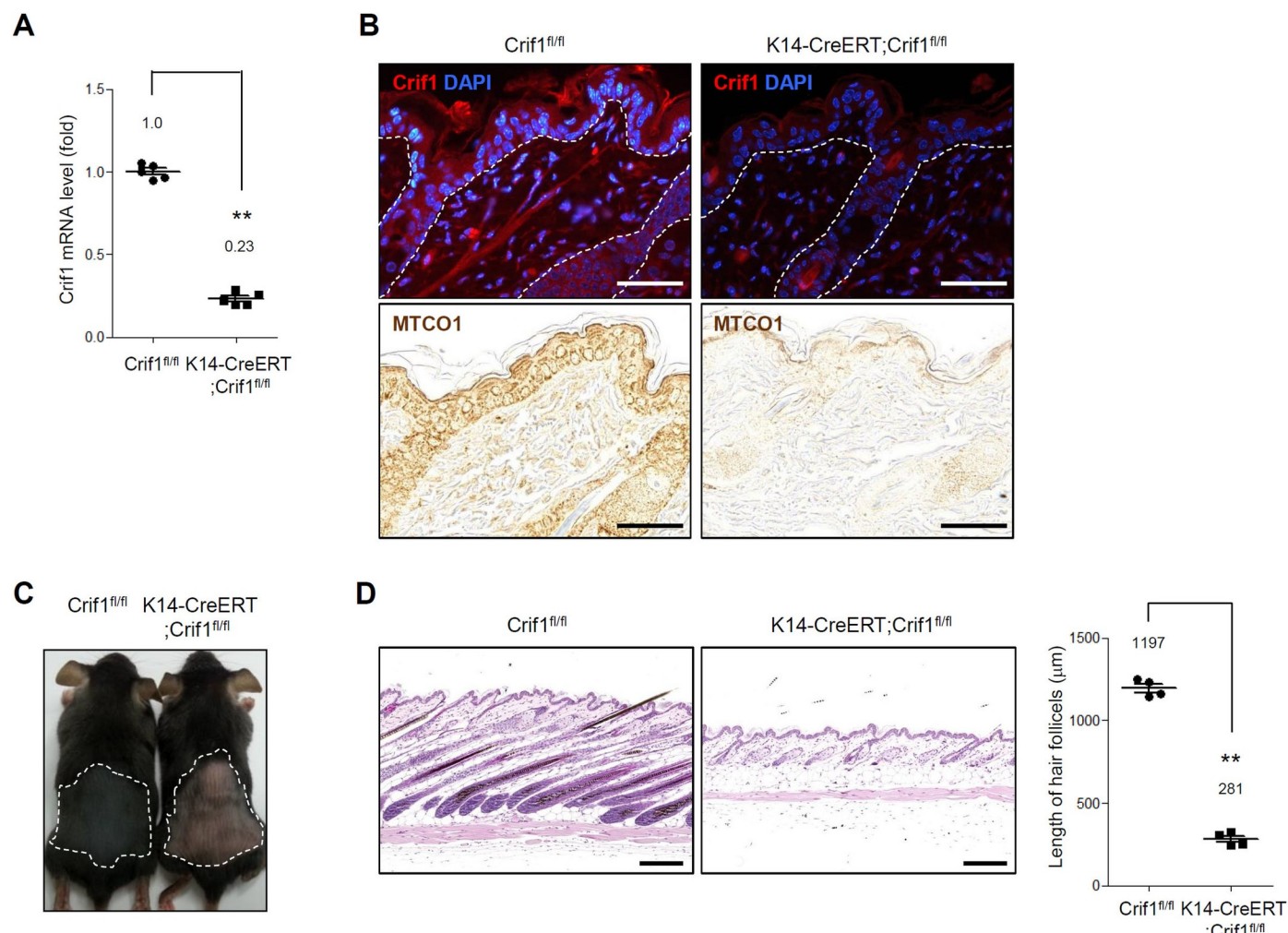

**Fig 1. Effect of Crif1 deficiency in epidermal cells.** Crif1$^{fl/fl}$ (WT) mice and K14-CreERT;Crif1$^{fl/fl}$ (Crif1 K14icKO) mice were shaved at P21 (telogen) and topically applied with 4-hydroxy tamoxifen (4-OHT) (1 mg/mice) for 5 days. Hair growth was examined at P35. (A) Total RNAs were isolated from epidermis at P35, and real time RT-PCR was performed (n = 4, $^{**}$P < 0.01). (B) Immunohistochemistry for Crif1 and MTCO1 in the epidermis of WT and Crif1 K14icKO mice at P35 (red, Crif1; blue, DAPI). A dotted white line indicates the basement membrane zone. Scale bar, 200 μm. (C) An image showing hair growth after 4-OHT treatment at P35. Anagen induction was delayed in Crif1 K14icKO mice compared to WT mice. (D) Histochemical analysis of skin sections from WT and Crif1 K14icKO mice at P35. Scale bar, 200 μm. Quantification of hair follicle length are represented in bar graph (n = 4, 10–15 hair follicles/mice, $^{**}$P < 0.01).

positive cells were detected in hair follicles of WT mice, whereas the Ki67 positive cells were barely detected in Crif1 K14icKO mice (S2 Fig). These data suggested that hair growth cycle was delayed in Crif1 K14icKO mice.

## Delayed hair growth cycle by Crif1 loss in hair follicle stem cells

During hair growth cycle, telogen-to-anagen transition is associated with activation of HFSCs [12]. To examine the role of Crif1 in HFSCs, we created HFSC specific icKO using K15-CrePR mice [13]. The K15-CrePR;Crif1$^{fl/fl}$ (Crif1 K15icKO) and littermate Crif1$^{fl/fl}$ (WT) mice were topically treated with RU486 from P21 for 5 days, and then hair growth was analyzed. Hair bulge-specific knockout of Crif1 was verified by immunohistochemical staining (S3 Fig). At P44, the back skins of WT and Crif1 K15icKO mice showed similar anagen appearances (Fig 2A). Histological analysis revealed that all the hair follicles were at fully grown anagen phase in

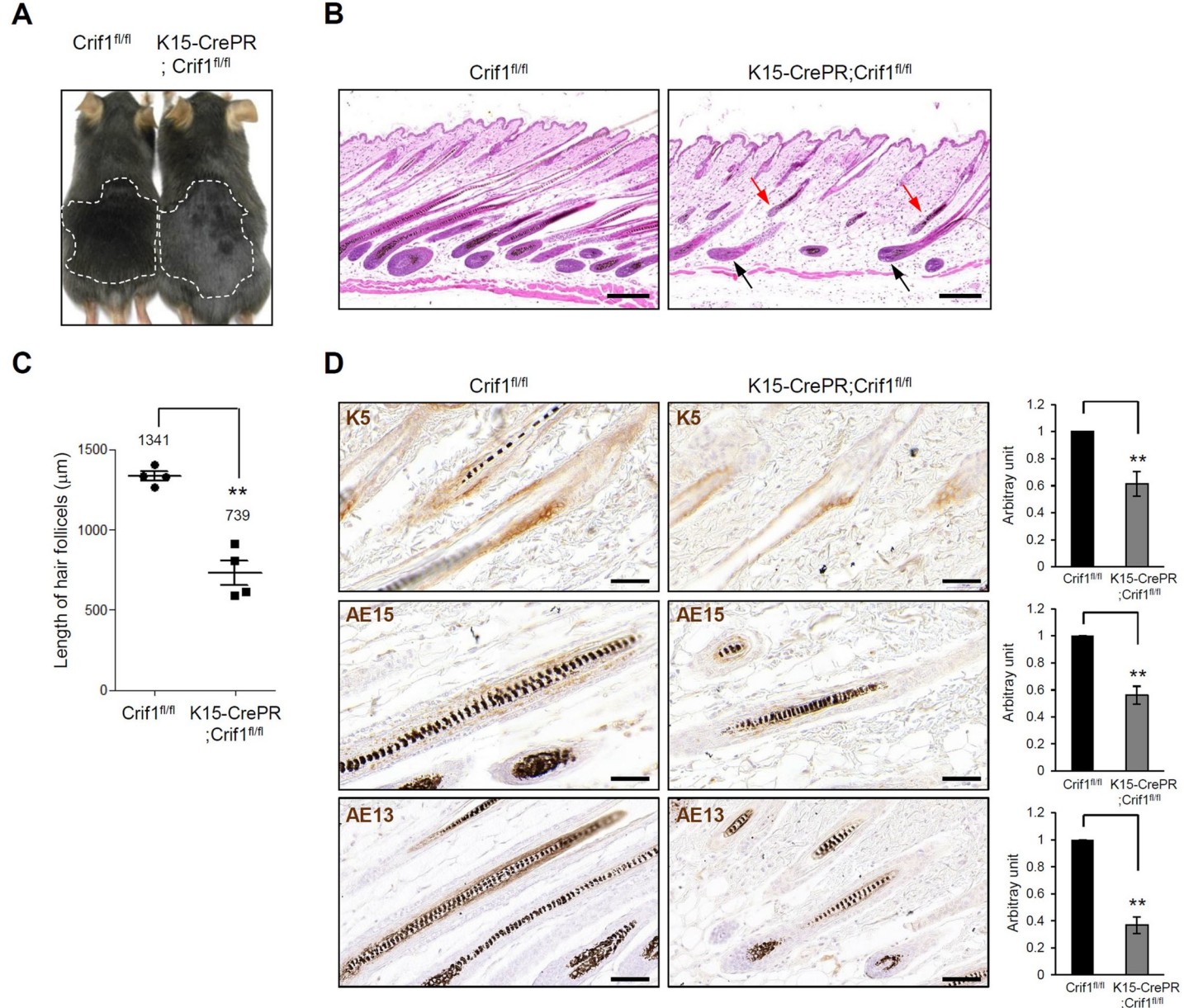

**Fig 2. Effect of Crif1 deficiency in hair follicle stem cells.** (A) Crif1^fl/fl (WT) mice and K15-CrePR;Crif1^fl/fl (Crif1 K15icKO) mice were shaved at P21 and then topically treated with RU486 (1 mg/mice) for 5 days. An image showing hair growth after RU486 induction at P44. (B) Histochemical analysis of skin sections from WT and Crif1 K15icKO mice at P44. In Crif1 K15icKO mice, some hair follicles were at fully grown anagen stage (black arrows), while some hair follicle were very tiny and did not reach to the subcutaneous fat layer (red arrows). Scale bar, 200 μm. (C) Quantification of hair follicle length are represented in bar graph (n = 3, 10–15 hair follicles/mice, **P < 0.01). (D) Immunohistochemistry for keratin 5 (K5), AE15, and AE13 in WT and Crif1 K15icKO mice at P44. Scale bar, 50 μm.

WT mice. However, in Crif1 K15icKO mice, some of hair follicles were at anagen stage (black arrows) but some of hair follicles were very tiny and did not grow to reach the lower fat layer (red arrows) (Fig 2B). Hair follicle length was significantly shorter in Crif1 K15icKO mice compared to WT mice (Fig 2C). Similar to Crif1 K14icKO mice, the Ki67 positive cells were significantly decreased in Crif1 K15icKO mice (S2 Fig). To investigate the effect of Crif1 knockout on the differentiation of hair follicle cells, we performed immunohistochemical staining using keratin 5 (K5), AE15 and AE13 antibodies to detect ORS, IRS, and hair cortex,

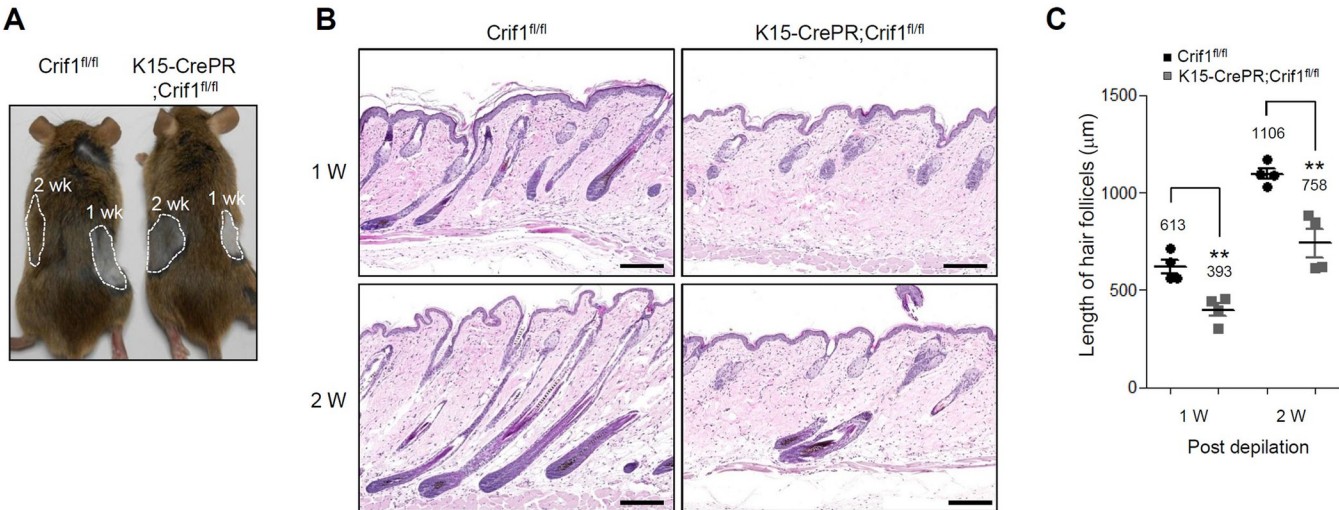

**Fig 3. Effect of Crif1 deficiency on depilation-induced anagen.** (A) Crif1$^{fl/fl}$ (WT) mice and K15-CrePR;Crif1$^{fl/fl}$ (Crif1 K15icKO) mice were intraperitoneally injected with RU486 (1 mg/mice) for 2 weeks (P35-P48) and then hair was plucked at P49 (2 weeks) and P56 (1 week). Hair growth was examined at P63. Compared to WT mice, depilation-induced anagen was retarded in Crif1 K15icKO mice. (B) H&E staining of skin sections from WT and Crif1 K15icKO mice at 1 and 2 weeks after depilation. Scale bar, 200 μm. (C) Quantification of hair follicle length are represented in bar graph (n = 3, 10–15 hair follicles/mice, **P < 0.01).

respectively. K5 immunoreactivity was observed in basal layer of ORS of both the WT and Crif1 K15icKO mice. As for AE15 and AE13 immunoreactivity, it was also weaker in Crif1 K15icKO mice than WT mice (Fig 2D). These results suggested that Crif1 loss affected the differentiation of hair follicles cells. We also examined the effect of Crif1 knockout on the differentiation of interfollicular epidermal cells. Immunohistochemical staining for differentiation markers, including loricrin and filaggrin, showed no difference between WT and Crif1 K15icKO mice (S4 Fig). These results indicated that conditional knockout of Crif1 in HFSCs did not affect the differentiation of interfollicular epidermal cells.

To further investigate the effect of Crif1 deficiency on hair growth cycle, we employed depilation-induced anagen model. [14] WT mice showed darker skin with new hair growth after depilation in a time-dependent manner, while Crif1 K15icKO mice showed brighter skin with delayed hair growth (Fig 3A). Histological analysis showed that WT mice displayed mid anagen at 1 week and late anagen at 2 weeks after depilation. However, Crif1 K15icKO mice showed early anagen at 1 week and mid anagen at 2 weeks after depilation (Fig 3B). Hair follicle length in Crif1 K15icKO mice was significantly shorter than that of WT mice, supporting the delayed hair growth cycle in Crif1 K15icKO mice (Fig 3C).

## Effect of Crif1 loss on maintenance of hair follicle stem cells

Despite the deficiency of Crif1 in HFSCs, hair growth cycle still progressed in Crif1 K15icKO mice. Thus, we checked whether Crif1 loss affected HFSC maintenance. In immunohistochemistry, K15-positive HFSCs were detected regardless of Crif1 deficiency in hair follicle bulge region of Crif1 K15icKO mice (Fig 4A). In addition, Lgr5-positive quiescent stem cells were also detected in both the WT and Crif1 K15icKO mice (S5 Fig). To characterize further the HFSC status, we carried out flow cytometry analysis using CD34 and integrin-α6 antibodies [15]. Results showed that CD34$^+$/integrin-α6$^+$ HFSCs were maintained in Crif1 K15icKO mice although the cell population was slightly decreased in Crif1 K15icKO mice compared to WT mice (Fig 4B). These data indicated that Crif1 was not required for maintenance of HFSCs.

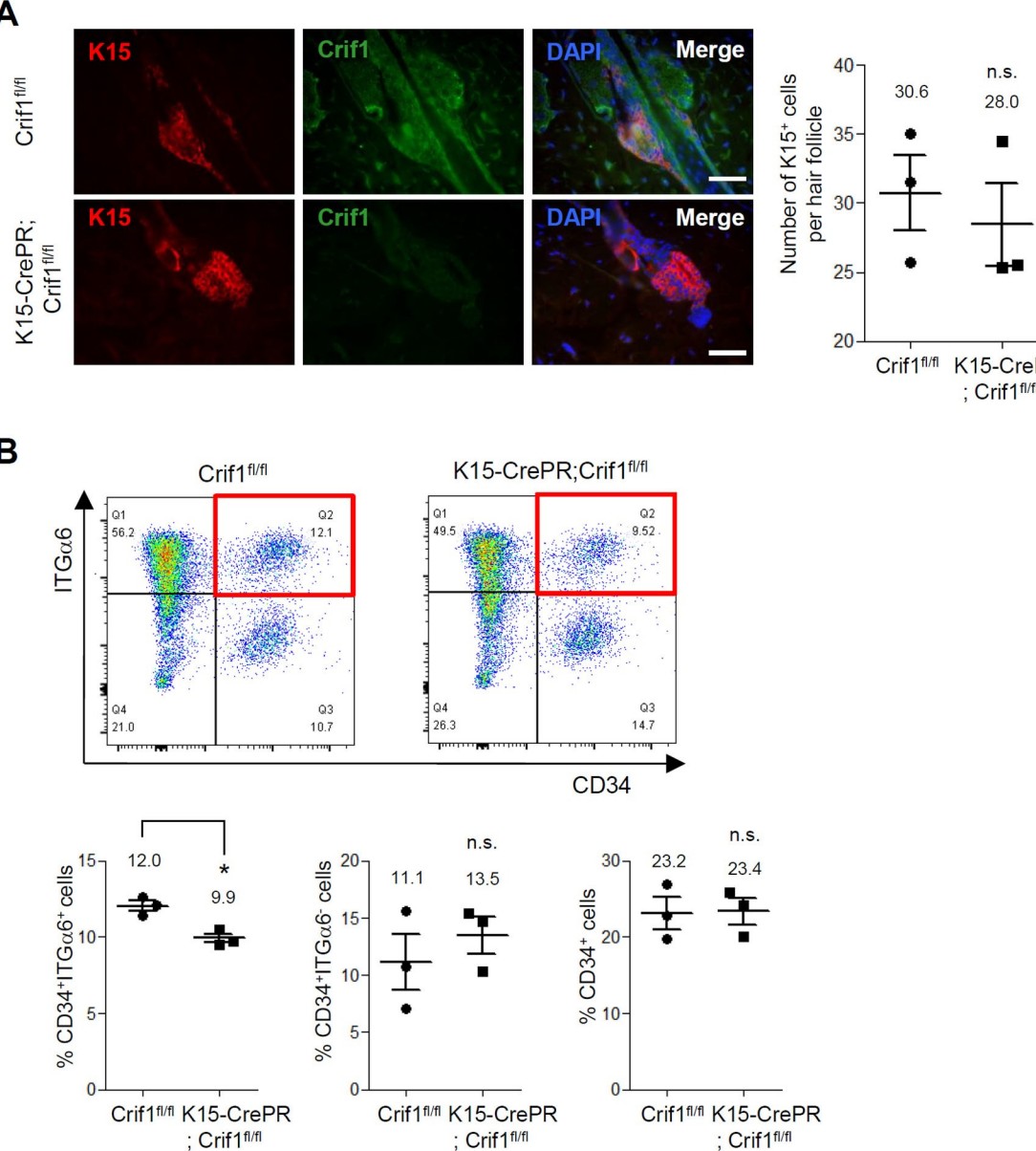

**Fig 4. Effect of Crif1 deficiency on maintenance of hair follicle stem cells.** (A) Crif1$^{fl/fl}$ (WT) mice and K15-CrePR;Crif1$^{fl/fl}$ (Crif1 K15icKO) mice were shaved at P21 and then topically treated with RU486 (1 mg/mice) for 5 days. Skin sections were prepared at P44 and double-stained with K15 (hair follicle stem cell marker) and Crif1 antibodies (red, K15; green, Crif1; blue, DAPI). Scale bar, 200 μm. The number of K15-positive cells per hair follicle was quantified (n = 3, n.s.: not significant). (B) Representative flow cytometry dot plots of epidermal cells labeled with CD34 and integrin-α6 antibodies. The CD34$^+$/integrin-α6$^+$ cells in WT and Crif1 K15icKO mice were highlighted in red box. Percentages of CD34$^+$/integrin-α6$^+$ cells (lower left), CD34$^+$/integrin-α6$^-$ cells (lower middle), and CD34$^+$ cells (lower right) are represented in bar graph (n = 3, $^*P < 0.05$).

## Discussion

Mitochondria are the important organelles that serve their essential role as the energy-producing center in the cells. In addition to this primary function, a variety of cellular activities are related to mitochondria. For example, imbalance of mitochondrial respiratory chain complexes in the epidermis results in severe inflammatory phenotype with massive immune cell infiltrates [16]. In other example, mitochondrial dysfunction leads to delayed wound closure

and reduced epidermal thickness together with epidermal stem cell exhaustion in older ages [17]. Although the majority of biological events that require energy expenditure are dependent on mitochondrial function, however various cellular events are still taken place in the absence of mitochondria-dependent energy production.

In this study, we examined the effect of mitochondrial dysfunction in terms of Crif1 deficiency on hair growth cycle in adult mice. Our data showed that inducible conditional knockout of Crif1 in epidermis (Crif1 K14icKO) and HFSCs (Crif1 K15icKO) retarded the hair growth cycle. In a previous study, Kloepper et al. has reported that intraepithelial ablation of electron transport chain (ETC) by deletion of mitochondrial transcription factor A (TFAM) delays morphogenesis of hair follicle, probably due to a significantly decreased proliferation rate of K14$^+$ cells. Furthermore, intraepithelial ETC ablation affects the epithelial-mesenchymal interactions, resulting in reduction of the inductive capacity of murine hair follicle mesenchyme [18]. Thus, it is possible that inducible conditional knockout of Crif1 in epidermis leads to similar defect in epithelial-mesenchymal interactions, thereby affecting the progression of hair growth cycle negatively. Elucidation of precise mechanism underlying retardation of hair growth cycle remains to be clarified.

In this study, Crif1 deficiency did not result in depletion of HFSCs in adult mice. Crif1 deletion using K15-CrePR system resulted in slight but significant reduction of CD34$^+$/integrin-α6$^+$ cell population (Fig 4B lower left graph). In contrast, CD34$^+$/integrin-α6$^-$ cell population increased slightly although there was no statistical significance (Fig 4B lower middle graph). Since it has been recognized that both the cell populations (CD34$^+$/integrin-α6$^+$ and CD34$^+$/integrin-α6$^-$) have same characteristics in terms of self-renewal and multipotency [19], eventually there was no difference in number of CD34$^+$ HFSCs between WT and Crif1 K15icKO mice (Fig 4B lower right graph). These results support the idea that mitochondria-dependent energy production is not required for HFSC maintenance. Since it has been well demonstrated that some stem cells including ESCs and iPSCs use the glycolysis to produce energy rather than mitochondrial OXPHOS system [20], it is plausible that similar metabolic event is employed by HFSCs. Based on the fact that HFSCs are relatively quiescent, it can be postulated that the glycolysis provides sufficient energy for maintenance of HFSC activity and Crif1 deficiency do not affect the HFSC maintenance critically. Elucidation of precise link between mitochondrial function and HFSC behavior will be an interesting further study.

In summary, we demonstrated that conditional knockout of Crif1 in epidermis and HFSCs retarded hair growth cycle, and that HFSC population was not affected by Crif1 deficiency. Our data suggest that mitochondrial function is important for the progression of hair growth cycle, but not for maintenance of HFSCs.

## Supporting information

**S1 Fig. Localization of Crif1 in mitochondria.** Cultured outer root sheath (ORS) cells were co-stained with mitotracker (red) and Crif1 (green). Nucleus was counterstained with 4,6-diamidino-2-phenylindole (DAPI, blue). Scale bar, 20 μm.
(TIF)

**S2 Fig. Immunohistochemical staining of proliferative cells in hair follicles.** (A) Crif1$^{fl/fl}$ (WT) mice and K14-CreERT;Crif1$^{fl/fl}$ (Crif1 K14icKO) mice were shaved at P21 and topically applied with 4-hydroxy tamoxifen (1 mg/mice) for 5 days. Skin sections were obtained at P35 and stained using Ki67 antibody. (B) Crif1$^{fl/fl}$ mice and K15-CrePR;Crif1$^{fl/fl}$ (Crif1 K15icKO) mice were shaved at P21 and topically treated with RU486 (1 mg/mice) for 5 days. Skin sections were obtained at P44 and stained using Ki67 antibody. Scale bar, 100 μm.
(TIF)

**S3 Fig. Crif1<sup>fl/fl</sup> (WT) mice and K15-CrePR;Crif1<sup>fl/fl</sup> (Crif1 K15icKO) mice were shaved at P21 and topically treated with RU486 (1 mg/mice) for 5 days.** Skin sections were obtained at P44 and stained using Crif1 antibody. Crif1 immunoreactivity was observed in epidermis of Crif1 K15icKO mice. In contrast, Crif1 immunoreactivity was very weak in hair bulge of Crif1 K15icKO mice. Scale bar, 100 μm (upper), 50 μm (lower).
(TIF)

**S4 Fig. Effect of Crif1 conditional knockout on epidermal differentiation.** Crif1<sup>fl/fl</sup> (WT) mice and K15-CrePR;Crif1<sup>fl/fl</sup> (Crif1 K15icKO) mice were shaved at P21 and topically treated with RU486 (1 mg/mice) for 5 days. Skin sections were obtained at P44 and stained using lori-crin (LOR) and filaggrin (FLG) antibodies. There was no difference in differentiation marker expression between WT and Crif1 K15icKO mice. Scale bar, 100 μm.
(TIF)

**S5 Fig. Crif1<sup>fl/fl</sup> (WT) mice and K15-CrePR;Crif1<sup>fl/fl</sup> (Crif1 K15icKO) mice were shaved at P21 and topically treated with RU486 (1 mg/mice) for 5 days.** Skin sections were obtained at P44 and stained using Lgr5 antibody. Lgr5 was detected in both the WT and Crif1 K15icKO mice. Scale bar, 50 μm.
(TIF)

**S6 Fig. Expression of Crif1 during hair cycle.** Scale bar, 200 μm.
(TIF)

## Author Contributions

**Conceptualization:** Chang-Deok Kim.

**Data curation:** Jung-Min Shin, Jung-Woo Ko, Chong-Won Choi, Chang-Deok Kim.

**Formal analysis:** Jung-Min Shin.

**Funding acquisition:** Chang-Deok Kim.

**Investigation:** Jung-Min Shin, Jung-Woo Ko, Chong-Won Choi, Chang-Deok Kim.

**Project administration:** Jung-Min Shin, Young-Joon Seo, Jeung-Hoon Lee, Chang-Deok Kim.

**Supervision:** Young Lee, Young-Joon Seo, Jeung-Hoon Lee, Chang-Deok Kim.

**Validation:** Chong-Won Choi, Young Lee.

**Writing – original draft:** Jung-Min Shin, Chang-Deok Kim.

**Writing – review & editing:** Jung-Min Shin, Young Lee, Chang-Deok Kim.

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
