## [Decision Letter · Decision Letter 0]

30 Sep 2019

PONE-D-19-23115

Deficiency of Crif1 in hair follicle stem cells retards hair growth cycle in adult mice

PLOS ONE

Dear Dr. Kim,

Thank you for submitting your manuscript to PLOS ONE. Your manuscript was reviewed by two external, expert, reviewers. Although the reviewers have indicated that your manuscript has merit, it does not fully meet PLOS ONE’s publication criteria in the current form. We invite you to submit a revised manuscript, if you so wish.  Please note that any revised manuscript will be resent to the same reviewers for further evaluation before a decision on publication could be rendered. It is therefore important that ALL points raised by the reviewers are addressed, adequately, in a revised manuscript.

We would appreciate receiving your revised manuscript by Nov 14 2019 11:59PM. To enhance the reproducibility of your results, we recommend that if applicable you deposit your laboratory protocols in protocols.io, where a protocol can be assigned its own identifier (DOI) such that it can be cited independently in the future. For instructions see: http://journals.plos.org/plosone/s/submission-guidelines#loc-laboratory-protocols

We look forward to receiving your revised manuscript.

Yours sincerely,

Aamir Ahmed

Academic Editor

PLOS ONE

Journal Requirements:

2. At this time, we request that you  please report additional details in your Methods section regarding animal care, as per our editorial guidelines: 1) Please provide details of animal welfare (e.g., shelter, food, water, environmental enrichment) 2) please describe any steps taken to minimize animal suffering and distress, such as by administering analgesics, 3) please include the method of sacrifice and 4) Please describe the post-operative care received by the animals, including the frequency of monitoring and the criteria used to assess animal health and well-being. Thank you for your attention to these requests."

3. For purposes of reporting, we request that you provide additional details as to the source of RU486 used in this study (please see

http://journals.plos.org/plosone/s/criteria-for-publication#loc-3 for more about our guidelines). Please provide the source, product number, the company from which it was purchased and a description of appearance.

'This study was supported by a grant from the National Research Foundation of Korea (NRF-

2017R1A2B4008810).'

'The funders had no role in study design, data collection and analysis, decision to

publish, or preparation of the manuscript.'

Additional Editor Comments (if provided):

Reviewers' comments:

Reviewer's Responses to Questions

Comments to the Author

1. Is the manuscript technically sound, and do the data support the conclusions?

Reviewer #1: Partly

Reviewer #2: Partly

2. Has the statistical analysis been performed appropriately and rigorously? 

Reviewer #1: Yes

Reviewer #2: Yes

3. Have the authors made all data underlying the findings in their manuscript fully available?

The PLOS Data policyrequires authors to make all data underlying the findings described in their manuscript fully available without restriction, with rare exception (please refer to the Data Availability Statement in the manuscript PDF file). The data should be provided as part of the manuscript or its supporting information, or deposited to a public repository. For example, in addition to summary statistics, the data points behind means, medians and variance measures should be available. If there are restrictions on publicly sharing data—e.g. participant privacy or use of data from a third party—those must be specified.

Reviewer #1: Yes

Reviewer #2: Yes

4. Is the manuscript presented in an intelligible fashion and written in standard English?

Reviewer #1: Yes

Reviewer #2: No

5. Review Comments to the Author

Reviewer #1: The manuscript by Jung-Min Shin et al. examined the effect of Crif1 genetic loss on hair follicle (HF). The authors showed that in adult skin epidermal specific ablation of Crif1 leads to a delayed anagen induction. A similar phenotype was observed when they ablated CRif1 in K15+ HF stem cells both in physiological telogen-to-anagen transition and also in depilation induced anagen. Finally, since a small reduction of HF stem cells in K15 specific Crif1 Knock out compared to control was found, the authors claimed that Crif1 is not required for maintenance of HF stem cell.

Although these observations are convincing and interesting, the authors should strengthen the manuscript with further analysis, since there are not any mechanistic insights.

Here some suggestions that should be taken in consideration to improve the manuscript:

1. In a previous paper from the same lab the authors showed the effect of of Crif1 epidermal ablation on the differentiation level of developing epidermal cells (Shin et al. Scientific Report 2017). Is differentiation affected in growing bulbs upon Crif1 loss? If so, there is any particularly affected lineage? The authors could check for some markers of differentiated lineage in the bulb (see for example fig1 in Kobielak et al. JCB 2003).

2. Reduced proliferation and increased apoptosis were previously detected in Crif1cKO developing epidermis (Shin et al. Scientific Report 2017). In the present inducible system, are HF stem cells less proliferation or a bit more apoptotic in the earlier stages of the telogen-to-anagen transition?

3. If a defect in differentiation, proliferation or apoptosis will be detected in HF cells (point 1 and 2), do interfollicular epidermal cells show a similar phenotype? If it is not the case, is this difference due to the fact that HF cells are more dependent on WNT signalling rather than interfollicular epidermal cells?

4. The authors claim that the level of MTCO1 is reduced in Crif1 K14icKO mice, indicating that mitochondrial dysfunction is achieved by Crif1 loss. However, the histology quality need to be improved (at least the image I can see in the pdf). Magnification and better quality images are needed since this is the only evidence for mitochondrial dysfunction. Is MTCO also decreased in the dermal cell as it looks from the image provided (fig.1 and 4)?

5. It would be very informative to have also a better staining for Crif1 to be able to see if in the adult epidermis Crif1 is present in the mitochondria/cytoplasm or in the nucleus (from literature, a crucial function in mitochondria but also Crif1 can have a transcriptional cofactor role (i.e. Stat3). This will help the interpretation of the result. In addition, a mitochondrial/cytoplasmic expression would support the use of epidermal Crif1cKO as a model to study mitochondrial dysfunction. While a nuclear staining would suggest that the phenotype observed in Crif1 cKO HFs might also be caused by a transcriptional deregulation.

6. In Velarde et al 2015, the authors showed that mitochondrial dysfunction can have opposite phenotype in skin regeneration. It would be interesting to know if the phenotype observed associated to a deficiency of Crif1 in hair follicle stem cells is age dependent.

7. Which statistical test has been used to calculate the p values?

Reviewer #2: This interesting work by Shin et al., show a link of the mitochondrial protein Crif1 with hair follicle cycling. The authors have previously published a study using constitutive K14-Cre driven Crif1 KO, which led to neonatal death. Whereas the current study focuses on inducible KO in adult mice to study hair cycle and hair follicle stem cell (HFSC) maintenance. The authors induce the deletion of Crif1 in the basal layer of epidermis (includes quiescent and proliferating stem cells and transit amplifying cells) using K14-CreERT2 mice and in the hair follicle bulge region (reservoir HFSC) using K15-CrePR mice. Both deletion models yield similar hair cycle phenotype – i.e., the progression of hair cycle is affected. The authors further analyzed the status of stem cell maintenance and found no obvious stem cell depletion in Crif1-deleted skin. While this study shed light on a crucial role of Crif1 in murine hair cycle, there are few concerns that need to be addressed before the publication of the manuscript in PLoS ONE.

To produce new hairs, existing follicles undergo cyclical bouts of growth (anagen), regression (catagen) and rest (telogen). HFSCs, especially the quiescent populations are activated during telogen to anagen transition phase to start a new round of hair growth. Therefore, hair cycle is a process that intricately dependent of HFSCs. So, it is puzzling why there is no effect of Crif1 deletion on HFSC maintenance despite there is a strong phenotype in hair cycle progression.

The data (Fig. 1, 2 and 3) regarding hair cycle is convincing but the stem cell maintenance part is not. In Fig. 4, the authors show that stem cell depletion is not observed. This conclusion needs further experimental support other than K15 staining and FACS analysis of CD34+ Itga6+ cells. A major concern is that the K15 expression doesn’t exclusively report the stem cell populations, rather it labels the whole bulge region which consists of quiescent and proliferative stem cell populations. If quiescent stem cell populations are affected, the authors wouldn’t observe it. It would be convincing if they show other bulge stem cell marker expression (such as Lgr5) either by immunohistochemistry or qPCR on CD34+ FACS sorted cells. They could also use CD34+ sorted cells to validate the deletion of Crif1 by qPCR as the immunostaining in Fig. 4A is not convincing. Crif1 staining is almost absent in all compartment of skin, one would expect to lose the expression in bulge region, not in other compartments. This need to be addressed.

In fact, an ideal and elegant experiment to show the HFSC maintenance is EdU-pulse chase to analyze the label retaining cells (LRC). However, this is a time-consuming experiment, which would normally take about 3 months to complete, hence I am not suggesting it here. Instead, the authors should consider showing proliferative state of the epidermis and hair follicles (in both inducible models) by Ki-67 immunostaining.

Other concerns:

1. Expression dynamics of WT Crif1 must be compared during anagen, catagen and telogen stages. This could be done using qPCR and immunohistochemistry. This data would help to understand the specificity of Crif1 function in hair cycle.

2. Fig. 1B – include a higher magnification image of interfollicular epidermis (IFE) expression of Crif1 in WT and icKO to show in which layers the Crif1 is expressed. Similarly, include images for MTCO1 expression as well.

3. Individual data points must be shown on the error bars of graphs.

4. In Fig. 1C and 2C – how many hair follicles per mouse was analysed? This information should be mentioned in the figure legends.

5. The manuscript would benefit from a minor grammatical revision.

6. PLOS authors have the option to publish the peer review history of their article (what does this mean?). If published, this will include your full peer review and any attached files.

Do you want your identity to be public for this peer review?For information about this choice, including consent withdrawal, please see our Privacy Policy.

Reviewer #1: No

Reviewer #2: No

---

## [Author Response · Author response to Decision Letter 0]

22 Dec 2019

Response to Reviewer

Reviewer #1: The manuscript by Jung-Min Shin et al. examined the effect of Crif1 genetic loss on hair follicle (HF). The authors showed that in adult skin epidermal specific ablation of Crif1 leads to a delayed anagen induction. A similar phenotype was observed when they ablated Crif1 in K15+ HF stem cells both in physiological telogen-to-anagen transition and also in depilation induced anagen. Finally, since a small reduction of HF stem cells in K15 specific Crif1 Knock out compared to control was found, the authors claimed that Crif1 is not required for maintenance of HF stem cell.

Although these observations are convincing and interesting, the authors should strengthen the manuscript with further analysis, since there are not any mechanistic insights.

Here some suggestions that should be taken in consideration to improve the manuscript:

1. In a previous paper from the same lab the authors showed the effect of Crif1 epidermal ablation on the differentiation level of developing epidermal cells (Shin et al. Scientific Report 2017). Is differentiation affected in growing bulbs upon Crif1 loss? If so, there is any particularly affected lineage? The authors could check for some markers of differentiated lineage in the bulb (see for example fig1 in Kobielak et al. JCB 2003).

 As you suggested, we performed immunohistochemical staining to determine the effect of Crif1 loss on differentiation in hair follicles. We used keratin 5 (K5) antibody for ORS and AE15 antibody for IRS. The K5 immunoreactivity was observed in basal layer of ORS of both the WT and Crif1 K15icKO mice. However, it was likely that K5 immunoreactivity in Crif1 K15icKO mice was slightly weaker than that of WT mice. As for AE15 immunoreactivity, it was also weaker in IRS of Crif1 K15icKO mice than WT mice. These results suggested that Crif1 loss affected the differentiation of epidermal cells in hair follicles. We added new data in revised Fig 2D. We also added the descriptions for Fig 2D in page 10 / line 2 and page 10 / line 22 of revised manuscript.

2. Reduced proliferation and increased apoptosis were previously detected in Crif1cKO developing epidermis (Shin et al. Scientific Report 2017). In the present inducible system, are HF stem cells less proliferation or a bit more apoptotic in the earlier stages of the telogen-to-anagen transition?

 We performed immunohistochemical staining using Ki67 antibody to examine the proliferation of hair follicle cells. The Ki67 positive cells were detected in the hair follicles of WT mice, whereas the Ki67 positive cells were barely detected in both the Crif1 K14icKO mice and Crif1 K15icKO mice (Supplementary Fig S2). When we performed TUNEL staining, we could not detect any difference between WT and inducible conditional knockout mice (data not shown). We added the description for Supplementary Fig S2 in page 8 / line 18 and page 9 / line 24 of revised manuscript.

3. If a defect in differentiation, proliferation or apoptosis will be detected in HF cells (point 1 and 2), do interfollicular epidermal cells show a similar phenotype? If it is not the case, is this difference due to the fact that HF cells are more dependent on WNT signalling rather than interfollicular epidermal cells?

 We examined whether conditional knockout of Crif1 affected the differentiation of interfollicular epidermis. Immunohistochemical staining for epidermal differentiation markers, including loricrin and filaggrin, showed no difference between WT and Crif1 K15icKO mice (Supplementary Fig S4). These results indicated that conditional knockout of Crif1 in HF stem cells did not affect the differentiation of interfollicular epidermal cells. As you pointed out, we think that there is a difference in cells' responsibility to extracellular signals such as WNT between WT and conditional knockout mice, and this may be a cause for difference in differentiation potential between hair follicle cells and interfollicular epidermal cells. The molecular basis underlying the difference between hair follicle cells and interfollicular epidermal cells remains to be disclosed. We added the description for Supplementary Fig S4 in page 10 / line 8 of revised manuscript.

4. The authors claim that the level of MTCO1 is reduced in Crif1 K14icKO mice, indicating that mitochondrial dysfunction is achieved by Crif1 loss. However, the histology quality need to be improved (at least the image I can see in the pdf). Magnification and better quality images are needed since this is the only evidence for mitochondrial dysfunction. Is MTCO also decreased in the dermal cell as it looks from the image provided (fig.1 and 4)?

 As you suggested, we performed MTCO1 immunostaining again. We changed Fig 1B with new staining data, which showed that expression of MTCO1 was significantly reduced in Crif1 K14icKO mice. As for dermal cell staining, it looked like MTCO1 also decreased as you pointed out. Actually, we performed all staining procedure simultaneously with WT and cKO paraffin blocks, and we did not know exactly the reason why it appeared less stained. We speculated that the difference in extracellular matrix deposition between WT and cKO mice affected the background staining.

5. It would be very informative to have also a better staining for Crif1 to be able to see if in the adult epidermis Crif1 is present in the mitochondria/cytoplasm or in the nucleus (from literature, a crucial function in mitochondria but also Crif1 can have a transcriptional cofactor role (i.e. Stat3). This will help the interpretation of the result. In addition, a mitochondrial/cytoplasmic expression would support the use of epidermal Crif1cKO as a model to study mitochondrial dysfunction. While a nuclear staining would suggest that the phenotype observed in Crif1 cKO HFs might also be caused by a transcriptional deregulation.

 We performed immunohistochemistry and changed Fig 1B, in which Crif1 is mainly detected in cytosol. To further examine the localization of Crif1, we also performed immunocytochemistry using mitotracker and Crif1 antibody in cultured ORS cells. As shown in Supplementary Fig S1, Crif1 co-localized with mitotracker in ORS cells. Thus we think that Crif1 localizes mainly in mitochondria. We added the words ‘Supplementary Fig S1’ in page 3 / line 19 of revised manuscript. 

6. In Velarde et al 2015, the authors showed that mitochondrial dysfunction can have opposite phenotype in skin regeneration. It would be interesting to know if the phenotype observed associated to a deficiency of Crif1 in hair follicle stem cells is age dependent.

 In Velarde's paper, Sod2 deficiency accelerated wound closure in young mice (4~8 month), whereas Sod2 deficiency delayed wound closure in old mice (11~14 month). In our experiment, we observed phenotype of Crif1 conditional knockout mice at less than 2 months old after birth. Thus, at the moment, we don't know if opposite phenotype will occur in old mice. The age-dependent phenotype change in Crif1 mice will be an interesting future study. 

7. Which statistical test has been used to calculate the p values?

 We statistically evaluated data using Mann-Whitney test. We added ‘statistical analysis’ in Materials and methods section (page 7 / line 14). 

Reviewer #2: This interesting work by Shin et al., show a link of the mitochondrial protein Crif1 with hair follicle cycling. The authors have previously published a study using constitutive K14-Cre driven Crif1 KO, which led to neonatal death. Whereas the current study focuses on inducible KO in adult mice to study hair cycle and hair follicle stem cell (HFSC) maintenance. The authors induce the deletion of Crif1 in the basal layer of epidermis (includes quiescent and proliferating stem cells and transit amplifying cells) using K14-CreERT2 mice and in the hair follicle bulge region (reservoir HFSC) using K15-CrePR mice. Both deletion models yield similar hair cycle phenotype – i.e., the progression of hair cycle is affected. The authors further analyzed the status of stem cell maintenance and found no obvious stem cell depletion in Crif1-deleted skin. While this study shed light on a crucial role of Crif1 in murine hair cycle, there are few concerns that need to be addressed before the publication of the manuscript in PLoS ONE.

To produce new hairs, existing follicles undergo cyclical bouts of growth (anagen), regression (catagen) and rest (telogen). HFSCs, especially the quiescent populations are activated during telogen to anagen transition phase to start a new round of hair growth. Therefore, hair cycle is a process that intricately dependent of HFSCs. So, it is puzzling why there is no effect of Crif1 deletion on HFSC maintenance despite there is a strong phenotype in hair cycle progression.

The data (Fig. 1, 2 and 3) regarding hair cycle is convincing but the stem cell maintenance part is not. In Fig. 4, the authors show that stem cell depletion is not observed. This conclusion needs further experimental support other than K15 staining and FACS analysis of CD34+ Itga6+ cells. A major concern is that the K15 expression doesn’t exclusively report the stem cell populations, rather it labels the whole bulge region which consists of quiescent and proliferative stem cell populations. If quiescent stem cell populations are affected, the authors wouldn’t observe it. It would be convincing if they show other bulge stem cell marker expression (such as Lgr5) either by immunohistochemistry or qPCR on CD34+ FACS sorted cells. They could also use CD34+ sorted cells to validate the deletion of Crif1 by qPCR as the immunostaining in Fig. 4A is not convincing. Crif1 staining is almost absent in all compartment of skin, one would expect to lose the expression in bulge region, not in other compartments. This need to be addressed.

 As you suggested, we purchased Lgr5 monoclonal antibody (OTI2A2) (Thermo Fisher Scientific, Cat# MA5-25644) and performed immunohistochemistry. As a result, Lgr5 was detected in both the WT and Crif1 K15icKO mice. This result indicated that quiescent stem cell populations were not affected by Crif1 loss. We added the description for Supplementary Fig S5 in page 11 / line 23 of revised manuscript.

 Regarding the Crif1 staining, we performed immunohistochemistry again using DAB staining method. As shown in Supplementary Fig S3, Crif1 immunoreactivity was observed in epidermis of Crif1 K15icKO mice. In contrast, Crif1 immunoreactivity was very weak in hair bulge of Crif1 K15icKO mice, indicating that bulge specific Crif1 knockout was achieved. We added the description for Supplementary Fig S3 in page 9 / line 18 of revised manuscript.

In fact, an ideal and elegant experiment to show the HFSC maintenance is EdU-pulse chase to analyze the label retaining cells (LRC). However, this is a time-consuming experiment, which would normally take about 3 months to complete, hence I am not suggesting it here. Instead, the authors should consider showing proliferative state of the epidermis and hair follicles (in both inducible models) by Ki-67 immunostaining.

 We performed immunohistochemical staining using Ki67 antibody to examine the proliferation of hair follicle cells. The Ki67 positive cells were detected in the hair follicles of WT mice, whereas the Ki67 positive cells were barely detected in both the Crif1 K14icKO mice and Crif1 K15icKO mice (Supplementary Fig S2). We added the description for Supplementary Fig S2 in page 8 / line 18 and page 9 / line 24 of revised manuscript.

Other concerns:

1. Expression dynamics of WT Crif1 must be compared during anagen, catagen and telogen stages. This could be done using qPCR and immunohistochemistry. This data would help to understand the specificity of Crif1 function in hair cycle.

 We performed immunohistochemical statining using Crif1 antibody during hair cycle of normal mice. As shown in Supplementary Fig S6, Crif1 was detected in almost all cells including epidermal keratinocytes and follicular cells. Particularly, Crif1 expression was higher in morphogenesis and anagen stage (P5-P15) than catagen (P19) and telogen (P21) stage. These results suggest that mitochondrial activity is increased in hair growing stage compared to regression and resting stages.

2. Fig. 1B – include a higher magnification image of interfollicular epidermis (IFE) expression of Crif1 in WT and icKO to show in which layers the Crif1 is expressed. Similarly, include images for MTCO1 expression as well.

 We performed immunohistochemisty and changed Fig 1B with new data. Crif1 and MTCO1 were expressed in all layers of epidermis.

3. Individual data points must be shown on the error bars of graphs.

 We added individual values in all graphs. (Fig 1A, Fig 1D, Fig 2C, Fig 3C, Fig 4A and Fig 4B).

4. In Fig. 1C and 2C – how many hair follicles per mouse was analysed? This information should be mentioned in the figure legends.

 We added the number of hair follicles used for analysis in the figure legends (page 9 / line 10, page 10 / line 22, page 11 / line 16)

5. The manuscript would benefit from a minor grammatical revision.

 We checked the grammar using word processor program.

---

## [Decision Letter · Decision Letter 1]

5 Feb 2020

PONE-D-19-23115R1

Deficiency of Crif1 in hair follicle stem cells retards hair growth cycle in adult mice

PLOS ONE

Dear Dr. Kim,

Thank you for submitting your manuscript to PLOS ONE. After careful consideration, we feel that it has merit but does not fully meet PLOS ONE’s publication criteria as it currently stands. Therefore, we invite you to submit a revised version of the manuscript that addresses the points raised during the review process.

Please pay specific attention to comments by reviewer no. 1. It would be helpful if you are able to conduct the proposed experiments in an expeditious manner. If not, could you please reply to the reviewers' concern in your rebuttal.

We would appreciate receiving your revised manuscript by Mar 21 2020 11:59PM. To enhance the reproducibility of your results, we recommend that if applicable you deposit your laboratory protocols in protocols.io, where a protocol can be assigned its own identifier (DOI) such that it can be cited independently in the future. For instructions see: http://journals.plos.org/plosone/s/submission-guidelines#loc-laboratory-protocols

We look forward to receiving your revised manuscript.

Kind regards,

Aamir Ahmed

Academic Editor

PLOS ONE

Reviewers' comments:

Reviewer's Responses to Questions

**Comments to the Author**

1. If the authors have adequately addressed your comments raised in a previous round of review and you feel that this manuscript is now acceptable for publication, you may indicate that here to bypass the “Comments to the Author” section, enter your conflict of interest statement in the “Confidential to Editor” section, and submit your "Accept" recommendation.

Reviewer #1: (No Response)

Reviewer #2: All comments have been addressed

2. Is the manuscript technically sound, and do the data support the conclusions?

Reviewer #1: Yes

Reviewer #2: Yes

3. Has the statistical analysis been performed appropriately and rigorously? 

Reviewer #1: Yes

Reviewer #2: Yes

4. Have the authors made all data underlying the findings in their manuscript fully available?

Reviewer #1: Yes

Reviewer #2: Yes

5. Is the manuscript presented in an intelligible fashion and written in standard English?

Reviewer #1: Yes

Reviewer #2: Yes

6. Review Comments to the Author

Reviewer #1: The authors have adequately addressed most comments raised in the previous round of review however there is still a point that I think it should be taken in consideration.

Since the Crif1 deletion does not lead to a loss of HF stem cells, while some defects of HF differentiation have been observed (AE15 staining), I think that the interpretation of the Crif1 inducible KO phenotype is linked to lower efficiency to produce differentiated HF progenies by the HF stem cells.

In this direction, only the AE15 staining has been performed. I suggest to add the staining for another HF differentiation marker and quantify the result from the two markers.

Minor point:

The sentence in the text about K5 is meaningless: ”However, it was likely that K5 immunoreactivity in Crif1 K15icKO mice was slightly weaker than that of WT mice ”.

Reviewer #2: The authors have addressed almost all the concerns I have raised in the first review. They performed experiments to prove that the quiescent cell populations were not affected by Crif1 deletion which further reinforces that Crif1 is dispensable for HFSC. Proliferation assay using Ki67 antibody further validates that the Crif1 deletion impedes the proliferation, thus contributing to hair cycle retardation likely by affecting proliferative stem cell population. The manuscript has been improved well and can be published in PLoS One upon minor revision on following;

Fig 4A – Add label for DAPI

Fig S2 – Very difficult to infer Ki67 expression from these images. Quantification must be done. Add label for Ki67 in each image.

Fig S3 – Area of magnification belongs to lower panel should be indicated in the upper panel.

Fig S4 – Add label for DAPI in each image.

Fig S5 – Add label for Lrig1 and DAPI in each image.

Fig S6 – Add label for Crif1 in each image and indicate its site of expression by arrow.

Individual data points in all graphs – I meant to include data points (dots) for each replicate, not the “cumulative value”. For example, if n=3 is carried out in an experiment, the distribution of each value represented as data points(=dots) should be overlaid on the error bar in a bar graph.

7. PLOS authors have the option to publish the peer review history of their article (what does this mean?). If published, this will include your full peer review and any attached files.

Reviewer #1: No

Reviewer #2: No

---

## [Author Response · Author response to Decision Letter 1]

18 Mar 2020

Response to Reviewer

Reviewer #1: The authors have adequately addressed most comments raised in the previous round of review however there is still a point that I think it should be taken in consideration.

Since the Crif1 deletion does not lead to a loss of HF stem cells, while some defects of HF differentiation have been observed (AE15 staining), I think that the interpretation of the Crif1 inducible KO phenotype is linked to lower efficiency to produce differentiated HF progenies by the HF stem cells.

In this direction, only the AE15 staining has been performed. I suggest to add the staining for another HF differentiation marker and quantify the result from the two markers.

As you suggested, we performed additional immunohistochemical staining using AE13 antibody obtained from Abcam (ab16113) to detect hair cortex. AE13 immunoreactivity in Crif1 K15icKO mice was markedly weaker than that of WT mice. This result supported that Crif1 deletion induces defects of hair follicle differentiation. We added new data in revised Fig 2D. We also added the descriptions for Fig 2D in page 10 / line 4, 6 and page 10 / line 23 of revised manuscript.

Minor point:

The sentence in the text about K5 is meaningless: ”However, it was likely that K5 immunoreactivity in Crif1 K15icKO mice was slightly weaker than that of WT mice ”.

We agreed with your point, and deleted the sentence in revised manuscript.

Reviewer #2: The authors have addressed almost all the concerns I have raised in the first review. They performed experiments to prove that the quiescent cell populations were not affected by Crif1 deletion which further reinforces that Crif1 is dispensable for HFSC. Proliferation assay using Ki67 antibody further validates that the Crif1 deletion impedes the proliferation, thus contributing to hair cycle retardation likely by affecting proliferative stem cell population. The manuscript has been improved well and can be published in PLoS One upon minor revision on following;

Fig 4A – Add label for DAPI

We added label for DAPI in Fig 4A. 

Fig S2 – Very difficult to infer Ki67 expression from these images. Quantification must be done. Add label for Ki67 in each image.

We quantified Ki67 positive cells and added graphs. We also added label for Ki67 in each image of Fig S2. 

Fig S3 – Area of magnification belongs to lower panel should be indicated in the upper panel.

We marked area of magnification in the upper panel of Fig S3.

Fig S4 – Add label for DAPI in each image.

We added label for DAPI in Fig S4.

Fig S5 – Add label for Lrig1 and DAPI in each image.

We added label for Lrig1 and DAPI in Fig S5.

Fig S6 – Add label for Crif1 in each image and indicate its site of expression by arrow.

We added label for Crif1 and indicated expression of Crif1 by arrow in Fig S6.

Individual data points in all graphs – I meant to include data points (dots) for each replicate, not the “cumulative value”. For example, if n=3 is carried out in an experiment, the distribution of each value represented as data points(=dots) should be overlaid on the error bar in a bar graph.

As your comments, we changed all graphs including data points. (Fig 1A, Fig1D, Fig 2C, Fig 3C, Fig 4A and Fig 4B)

---

## [Editor Report · Decision Letter 2]

10 Apr 2020

Deficiency of Crif1 in hair follicle stem cells retards hair growth cycle in adult mice

PONE-D-19-23115R2

Dear Dr. Kim,

We are pleased to inform you that your manuscript has been judged scientifically suitable for publication and will be formally accepted for publication once it complies with all outstanding technical requirements.

With kind regards,

Aamir Ahmed

Academic Editor

PLOS ONE
---

## [Editor Report · Acceptance letter]

14 Apr 2020

PONE-D-19-23115R2 

Deficiency of Crif1 in hair follicle stem cells retards hair growth cycle in adult mice 

Dear Dr. Kim:

I am pleased to inform you that your manuscript has been deemed suitable for publication in PLOS ONE. Congratulations! Your manuscript is now with our production department. 

With kind regards,

on behalf of

Dr. Aamir Ahmed 

Academic Editor

PLOS ONE